# Local adaptation and future climate vulnerability in a wild rodent

Silvia Marková[1], Hayley C. Lanier [2,3], Marco A. Escalante[1], Marcos O. R. da Cruz[2,3], Michaela Horníková [1], Mateusz Konczal [4], Lawrence J. Weider[2], Jeremy B. Searle [5] & Petr Kotlík [1] ✉

As climate change continues, species pushed outside their physiological tolerance limits must adapt or face extinction. When change is rapid, adaptation will largely harness ancestral variation, making the availability and characteristics of that variation of critical importance. Here, we used whole-genome sequencing and genetic-environment association analyses to identify adaptive variation and its significance in the context of future climates in a small Palearctic mammal, the bank vole (*Clethrionomys glareolus*). We found that peripheral populations of bank vole in Britain are already at the extreme bounds of potential genetic adaptation and may require an influx of adaptive variation in order to respond. Analyses of adaptive loci suggest regional differences in climate variables select for variants that influence patterns of population adaptive resilience, including genes associated with antioxidant defense, and support a pattern of thermal/hypoxic cross-adaptation. Our findings indicate that understanding potential shifts in genomic composition in response to climate change may be key to predicting species' fate under future climates.

As global climate continues to diverge from historical norms, species increasingly face conditions outside their physiological tolerance limits to which they must adapt or face extinction[1,2]. Because the emergence and spread of new mutations can be slow, adaptation often occurs through the redistribution of existing ancestral variation[3], which can be more rapidly acted on by environmental change[4,5]. Therefore, the probability of avoiding demographic collapse or extinction largely depends on diversity within a species and whether populations that are already adapted to different climate extremes can provide those alleles to populations facing similar environmental stressors in the future[6]. Recent genomic studies have confirmed the role of existing (i.e., standing) intraspecific variation in adaptation for some species[7–9]. However, for most species, the availability of adaptive variation and the potential for it to be accessible for future adaptation is largely unknown. The lack of relevant variation may be a limiting

factor for species that need to cope with the pace of human-induced rapid climate change.

One potentially important factor shaping the distribution of variation is historical cycles of divergence and subsequent gene flow between previously isolated populations, which typically increase standing variation[5]. These scenarios are common for temperate species whose populations have experienced recurrent waves of extirpation and range expansion driven by glacial-interglacial cycles and can facilitate adaptation to new local environments[5,10–12]. For example, the allelic composition of populations colonizing newly available habitats at the end of the last glaciation was often highly heterogeneous, containing genetic variants arising in multiple glacial refugia[13–17]. This variation likely provided a useful pool of adaptive alleles to facilitate local adaptation to the diverse environmental conditions that populations encountered during range expansion, but may also be relevant

[1]Laboratory of Molecular Ecology, Institute of Animal Physiology and Genetics of the Czech Academy of Sciences, Rumburská 89, 277 21 Liběchov, Czech Republic. [2]School of Biological Sciences, University of Oklahoma, 730 Van Vleet Oval, Norman, OK 73019, USA. [3]Sam Noble Museum, University of Oklahoma, 2401 Chautauqua Ave, Norman, OK 73072, USA. [4]Faculty of Biology, Evolutionary Biology Group, Adam Mickiewicz University, Poznań, Poland. [5]Department of Ecology and Evolutionary Biology, Corson Hall, Cornell University, Ithaca, NY 14853, USA. ✉e-mail: kotlik@iapg.cas.cz

to future climate change. Species whose populations evolved from lineages of different glacial origins and now inhabit geographic (e.g., latitudinal) environmental gradients may therefore offer particularly promising conditions for studying the role of standing variation in local adaptation and resiliency to future climate change.

Here we examine adaptive variation and its significance in the context of future climate change in the Celtic fringe, a remarkably human-like geographic pattern of genetic variation that emerged from the end-glacial colonization of Britain by small mammals[18]. Small mammals, including voles and shrews, colonized Britain while it was still connected to continental Europe at the end of the last glacial period, in a two-stage process from two distinct glacial refugia for each species[18]. Based on a range of genetic markers in modern[18,19] and ancient samples[20], this process involved a partial displacement of the first wave of colonists by a second after the decline of the first population during the Younger Dryas, with the original group confined to peripheral regions of Britain (the Celtic fringe areas, especially Scotland) while being displaced in areas of Britain closer to continental Europe (especially England). However, whether the resulting distribution of genetic variation observed in these species shapes their local adaptation remains an unresolved question. Mixed ancestry combined with a gradient of climatically-driven selection pressure in Britain[21], manifested as a transition from a relatively wet and cool periphery to a drier and warmer continent-adjacent area[22], makes the species with Celtic fringes excellent model systems to study the effects of standing genetic variation segregating in natural populations on rapid adaptation to new climates.

We focus on one of the species, the bank vole, *Clethrionomys glareolus* (also known as *Myodes glareolus*[23]). In the bank vole, admixture of the two colonist waves is associated with a gradient of genomic proportions of two distinct ancestors, with populations in the north (mainly Scotland) largely descended from the earliest colonists, while populations in the south (England and Wales) descended from the second colonization[19]. The bank vole is an excellent model for studying local adaptation in the Celtic fringe and in admixed populations in general. Not only is it an emerging model species representing forest ecosystems and potentially providing information on a wide range of species colonizing similar habitats, but it also exhibits polymorphism in hemoglobin (Hb), with differences between the two colonists[24,25] shaping evolutionary potential for populations under future climate change[26]. Bank vole Hb appears subject to climate-driven selection, with two distinct alleles with functional and physiological differences restricted to the colder northern and warmer southern parts of Britain, respectively[25,26]. This variation is likely the result of selection on a charge-changing amino acid substitution (S52C) in the β-subunit that significantly increases erythrocyte resistance to oxidative stress[25], an important source of selection pressure induced by environmental stressors such as extreme temperatures that affect survival and longevity of organisms under climate change[6,27,28].

Predicting organismal responses to climate necessitates moving beyond individual genes to detect signatures of local adaptation on a genome-wide scale. This will allow us to investigate the genetic basis of complex adaptive traits, such as resistance to climate-induced stress, and model their distribution in the future. While methods that identify targets of divergent selection relative to background differentiation are one important approach[29], genetic-environment association (GEA) approaches offer complementary insights into allele frequencies correlated with climatic variability. These approaches are powerful for detecting loci involved in climate adaptation, particularly when selective forces at individual loci are weak—which is likely for complex traits—and for postulating sources of selection pressure[30,31]. Importantly, GEA methods allow the separation of the respective effects of drift and selection in generating and maintaining variability by controlling for patterns of neutral population structure[32,33]. Once locally adaptive variation is identified, we can quantify the genomic offset of a population—i.e., the magnitude of change in the genetic composition of a population required to track future environmental conditions[34–36]. This provides insights into the risk of disrupting local adaptation (i.e., maladaptation) and the availability of adaptive variation needed to reverse disruptions[32,33]. This both advances our knowledge of local adaptation and provides insights into whether and how populations can survive in the context of future climates, which is critical for developing sound conservation and management strategies to address global biodiversity loss.

By whole-genome resequencing of 111 bank voles, we identify genes and biological processes important for local adaptation, use future climate models to predict maladaptation across Britain and assess the availability of standing variation required by populations for future adaptation[32,33]. Britain, although only a subset of the bank vole range, represents an ideal microcosm for the study of genetic-environment associations in this species, combining a broad range of environments with a relatively simple biogeographic history for the species[18,19]. Results suggest that regional differences in climate select for traits that influence patterns of population resilience, including genes associated with antioxidant defense. Populations in southern Britain are predicted to have the greatest relative chance of surviving future climate change with standing variation, whereas populations in the north (Scotland) are likely to require an influx of adaptive variation from southern populations, paralleling previous results for Hb[26]. The concordance between genes associated with high temperatures and genes associated with high altitudes in other mammals may indicate a reciprocal adaptation to thermal and hypoxic conditions.

## Results

Genomic variation among populations of bank voles in Britain (Supplementary Tables 1 and 2) follows a geographic gradient, with the first and second principal component axes separating the most geographically distant and genomically divergent populations (Supplementary Fig. 1). This is consistent with population structure identified by Admixture, which revealed an ancestry gradient at $K = 2$ (Fig. 1) consistent with two-stage colonization from different glacial refugia, with the first wave of colonists partially displaced by a second[18,19]. At $K = 3$, the best fit for this dataset, the northernmost site in Scotland (ABD) separated from the northern cluster while still maintaining a high degree of shared ancestry with the site in southern Scotland (MLN), as expected under a two-phase colonization with ongoing isolation-by-distance (Fig. 1).

### Genomic signatures of local adaptation

We used a combination of three population differentiation and GEA methods to detect signatures of divergent selection[37] (Fig. 2). The $F_{ST}$ outlier approach (the 95th percentile) and pcadapt ($q$-value, FDR < 0.1) detected 12,055 and 13,387 candidate SNPs, respectively (see Supplementary Table 3). The third method, partial redundancy analysis (pRDA), identifies SNPs with unusually strong correlations between allele frequencies and environmental variables, suggesting a selection advantage in a particular environment[37]. Bioclimatic variables retained by the pRDA were annual mean temperature (BIO 01), isothermality (BIO 03), maximum temperature of the warmest month (BIO 05), and annual precipitation (BIO 12; Supplementary Fig. 2). The full model was highly significant and explained 23% of the total genetic variance. Climate remained a strongly significant driver of genetic variance patterning, even when controlling for the effects of population structure and geography, explaining most of the variation (27%) explained by the full model (Supplementary Table 4). The pRDA method identified 20,383 outliers ($q$-value, FDR < 0.1). A total of 1075 SNPs (hereafter candidate loci)

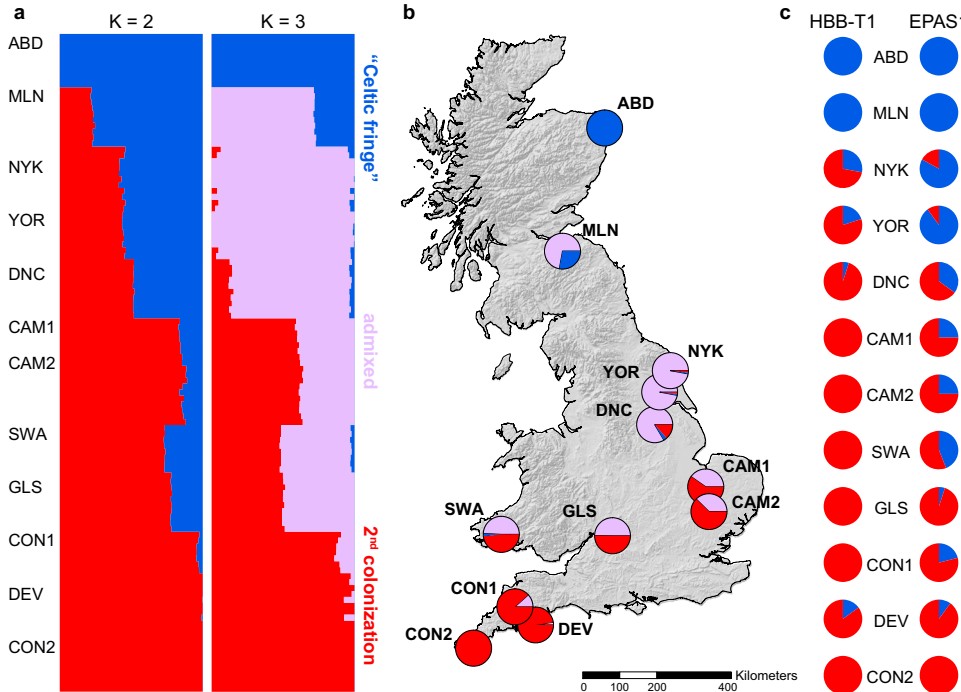

**Fig. 1 | Population structure of bank voles in Britain. a** Genome-wide ancestry proportions at each of the 12 sampling sites estimated with Admixture for the number of clusters $K = 2$ (consistent with two-stage colonization from different glacial refugia) and $K = 3$ (best fit for the dataset); **b** map showing admixture proportion for $K = 3$ per each site; **c** covariance in allele frequencies of selected candidate adaptive loci associated with climate adaptation. Population codes refer to Supplementary Table 1. The map was created using ArcMap (v. 10.8) with the Esri World Countries dataset (https://www.esri.com) and SRTM elevation data from WorldClim (https://www.worldclim.org/data/worldclim21.html).

overlapped between all three approaches (Fig. 2b; Supplementary Table 3). These loci are associated with climate, show evidence for divergent selection in bank voles, and primarily follow a north-south gradient, with higher fixation at latitudinal extremes (Supplementary Fig. 3).

### Understanding the adaptive landscape

To evaluate the geographic distribution of adaptive variation, we subjected 13,910 climate-related adaptive loci identified in the pRDA ($q$-value, FDR < 0.05; Supplementary Table 3) to a second, adaptively enriched pRDA[32,38] and identified three main gradients of bank vole adaptation to climate in Britain. These first three RDA axes (Supplementary Fig. 2) together explained 98% of the adaptive genetic variance among populations. The first axis (RDA1; 45% of variance) was strongly associated with high maximum temperature of the warmest month (positive RDA1 scores) and high annual precipitation (negative RDA1 scores), whereas RDA2 (28%) contrasted high isothermality and high annual precipitation with high maximum temperature of the warmest month (Supplementary Fig. 2a). Projection of these adaptive gradients onto the landscape (Fig. 3a) showed a contrast in adaptive index between south-eastern (essentially England) and north-western Britain (generally Scotland and Wales), suggesting that south-eastern populations are adapted to warmer temperatures and drier climates than north-western populations. In particular, northwest Scotland is estimated to be at the extreme of the gradient of adaptation to cold and wet climates (lowest RDA1 index), but this should be interpreted with caution because we did not sample this area and the extrapolation may not accurately reflect genomic differentiation in this region. In contrast, RDA3 (21%) associated primarily with isothermality (negative RDA3 scores; Supplementary Fig. 2b), distinguished coastal and southern areas characterized by adaptation to lower intra-annual temperature fluctuation from upland areas (Fig. 3a).

### Risk of future maladaptation and availability of adaptive variation

To predict how the geographic distribution of adaptive variation impacts future population survival, we extrapolated the genetic-environment relationship derived from pRDA to future environments to estimate the genomic and geographic offsets[32,33] (Fig. 4). For comparison, we also estimated genomic and geographic offsets using gradient forest (GF)[34]. The offsets derived from pRDA and GF were very similar, so we focus on pRDA in the main text but provide the GF results in Supplementary Fig. 4. The extrapolation of genomic offset revealed a contrasting pattern (Fig. 4a), with higher maladaptation risk predicted in central England (genomic offset 1.5–2.4), with populations in Scotland and the southwestern coast at lower risk (genomic offset 1–1.5). In contrast, the highest values for geographic offset were found in northern Scotland and the lowest in eastern England, suggesting that populations in Scotland are physically furthest from those with an adaptive composition that matches future climates, whereas populations in eastern England are closest (Fig. 4b).

When considering current adaptive genetic variation (Fig. 4c), populations in Scotland, at the northern end of the admixture gradient, and southwest England, at the southern end, had combinations of low levels of standing genetic variation (SGV) and high population adaptive indexes (PAIs). In contrast, populations in central England and Wales, in the middle of the gradient, had high levels of SGV and low PAIs. Finally, populations in eastern England had moderate levels of SGV but high PAIs. In all populations, SGV was slightly higher at adaptive SNPs than at putative neutral SNPs (Supplementary Fig. 5).

### Candidate genes for adaptation to climate

Of the 1075 candidate SNPs, 172 SNPs were located in coding and flanking regions; 126 SNPs were matched to mouse genes and used in gene ontology analyses (Supplementary Data 1). These indicate a significant functional enrichment of 15 biological processes, 14 cell

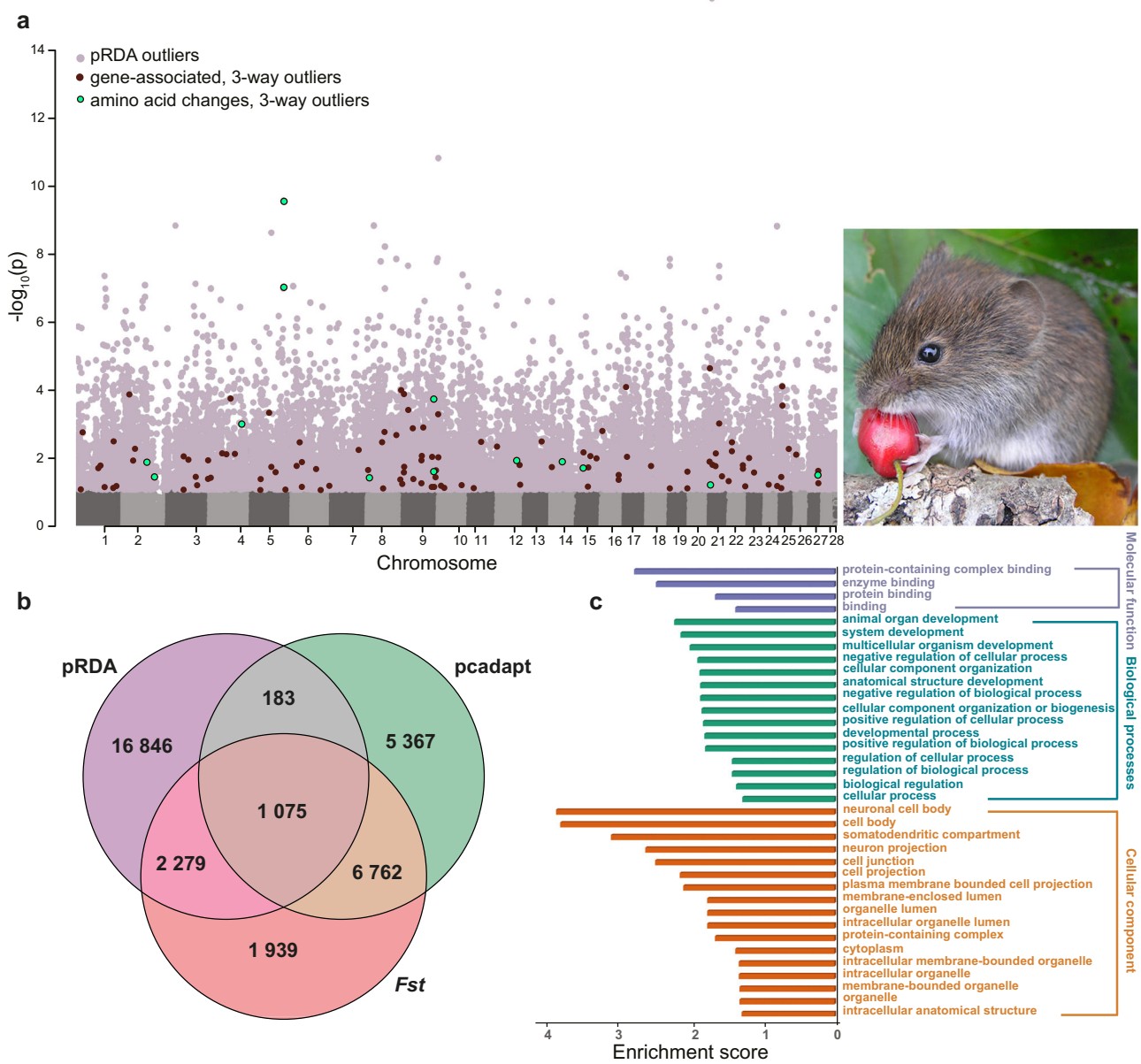

**Fig. 2 | Candidate adaptive loci. a** Manhattan plot of partial RDA (pRDA) outlier SNPs; **b** Venn diagram of the number of outliers identified by three methods (pRDA, pcadapt, and $F_{st}$); and **c** gene ontology classification of candidate genes by molecular function, biological process, and cellular component.

components and four molecular function categories (Fig. 2). Candidate loci are associated with climatic gradients along all three main RDA axes (Fig. 3). There were 19 SNPs from candidate loci that resulted in amino acid changes; 13 of these SNPs matched mouse protein-coding genes. Two amino acid changes were detected in both the *Arhgap32* (regulates the actin cytoskeleton) and the *Eftud2* (role in the pre-mRNA splicing process) genes and another 9 adaptive SNPs caused only one amino acid change (Supplementary Data 1). Adaptive candidates included genes involved in response to oxidative stress (GO:0006979; *Adprs*, *Dhcr24*, *Epas1*, *Prkca*, *Prkn*, and *Txn2*). Another GO category of interest included actin binding (GO:0003779; *Hcls1*, *Prkn*, *Sptbn5* and *Tln1*). Additionally, two genes, *HBB-T1* and *Epas1*, frequently implicated in the response to hypoxia together with another 16 hypoxia-linked genes accounted for approximately 16% of all candidate loci (Supplementary Data 1).

## Discussion

Here, we provide evidence for local adaptation to climate in bank voles in Britain. Population structure there was strongly shaped by the two-phase colonization process, with rapid population expansion and admixture after the Younger Dryas[18,19] and recent isolation-by-distance shaping the distribution of variation (Fig. 1a, b). While genetic variation in bank voles in Britain was influenced by geographic distance and demographic history, climate was an important driver of adaptive variation there, underscoring the role of local conditions in shaping genetic composition. Analyses indicate that local adaptation involves many genes linked to cellular and physiological processes. While the highly admixed west-central populations are genomically best prepared for adaptation to future climatic conditions (although with greatest need for adaptive genetic change), the non-admixed populations at the northern extremes (i.e., the Celtic fringe) may require an influx of adaptive variation to adapt in the future. By modeling adaptive potential along this climatic gradient, these results allow us to better predict how relative resilience and vulnerability to climate change are distributed geographically and mediated across the genome.

Our results indicate multilocus selection−i.e., selection on many loci of small effect−in response to environmental gradients has shaped

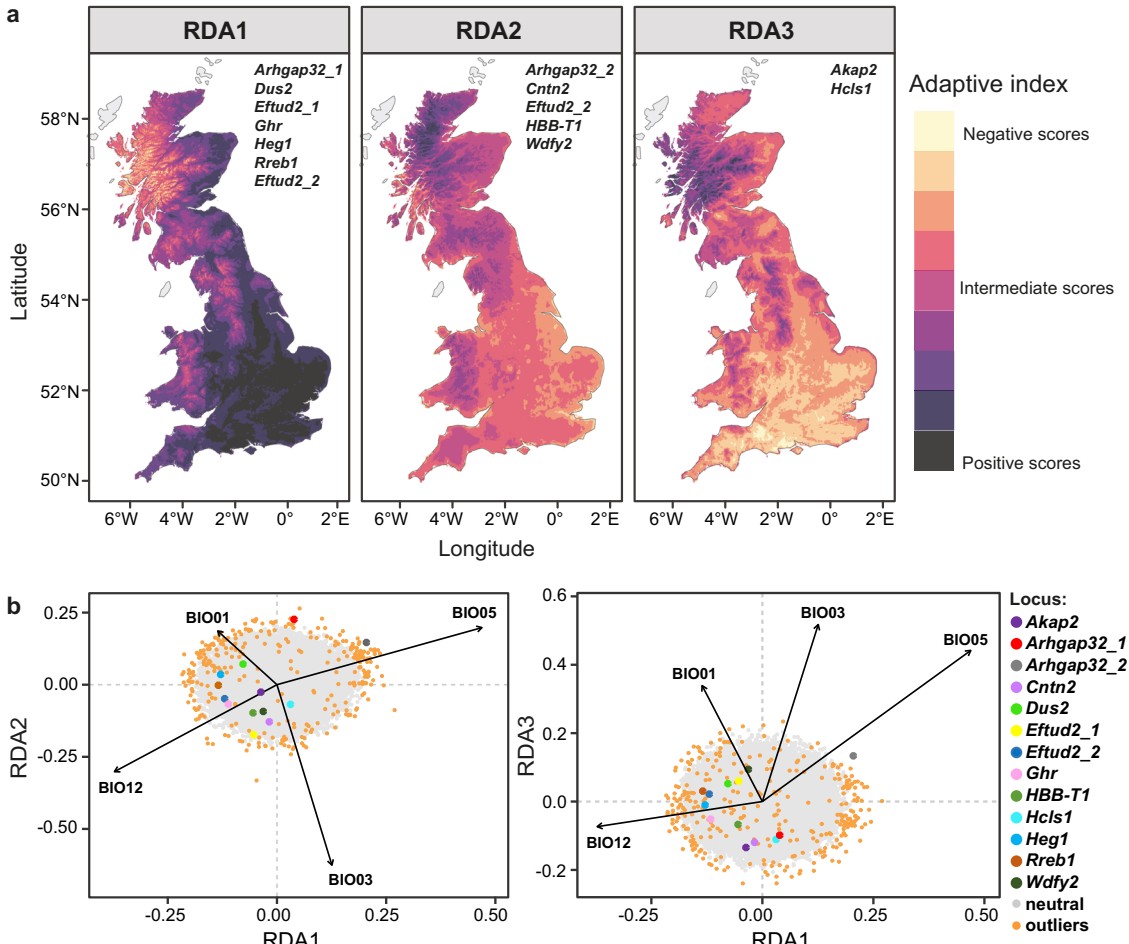

**Fig. 3 | Local adaptation in British bank voles. a** Spatial projection of adaptive genetic turnover across the bank vole range in Britain and the list of 13 candidate genes with amino acid changes associated with each pRDA axis (RDA1–RDA3). Two amino acid changes were detected for *Arhgap32* and *Eftud2*, whereas only one amino acid change was documented for all other genes. **b** RDA plot showing the association between the candidate adaptive loci and the four bioclimatic variables BIO 01 (annual mean temperature), BIO 03 (isothermality), BIO 05 (maximum temperature of the warmest month), and BIO 12 (annual precipitation) in the adaptively enriched genetic space.

adaptive capacity in populations across Britain[32,39]. In bank voles, adaptation occurs along two main axes that separate populations in the southeast and northwest of Britain, with south-eastern populations better adapted to warmer temperatures and drier climates than northwestern populations (Fig. 3a). Most of the remaining variation distinguishes coastal and southern areas along the third RDA axis, with lower intra-annual temperature variability, from upland regions (Fig. 3a). Allele frequencies of candidate SNPs largely follow these major gradients (Supplementary Fig. 3), with many alleles fixed at the northern end of the distribution, i.e., the Celtic fringe. However, loci with a lack of fixation at the northern extreme may indicate gene flow from the south, as has been predicted for hemoglobin[26]. For other SNPs, the same allele is nearly or entirely fixed at both extremes (Supplementary Fig. 3 and Table 5), which may indicate a function beneficial outside the thermal neutral zone[40]. Future climatic models predict an increased frequency of extreme temperature anomalies, as well as higher average temperatures and rainfall[41], which will continue to drive selection along these axes.

Given the rapid pace of climate change, adaptation must necessarily rely on pre-existing genetic variation (e.g., standing genetic variation) within and among populations[7–9]. Our models predict increased climate vulnerability for populations in central England (i.e., the largest genomic offset), but there may not be a great risk of maladaptation for these populations because they are relatively close to sources of suitable adaptive alleles (i.e., a small geographic offset).

Although the predicted genomic offset is lower for the two northern-most populations (i.e., the Celtic fringe) and those in southwest England, they are already at the extreme bounds of potential genetic adaptation (high population adaptive indices, PAI; high percentage of fixed alleles) and exhibit low standing genetic variation (SGV). When PAI is very high and most adaptive alleles are already or nearly fixed, populations have little potential for future adaptation with the standing variation they possess[32]. Because climate adaptation is often a polygenic process[31], maintaining variability at adaptive loci may increase population viability under future environmental changes. The importance of SGV in local adaptation is supported by the higher SGV at adaptive SNPs compared to putatively neutral SNPs in all populations. Populations lacking adaptive variability may not be able to compensate for climate change over the next few decades without the influx of adaptive variation from other populations. Our results suggest that migration from central England and Wales may provide adaptive genetic material to these Celtic fringe populations; however, the distance to populations with suitable adaptive variants is great (i.e., there is a large geographic offset). In addition to the distance that alleles must travel, adaptation through gene flow also depends on population connectivity, and habitat fragmentation may prevent the migration of adaptive alleles even if they occur nearby. The bank vole is common in brushy edge habitats in Britain, and its population structure is not particularly high for a small mammal, suggesting that there are no strong geographic barriers and that the spread of adaptive

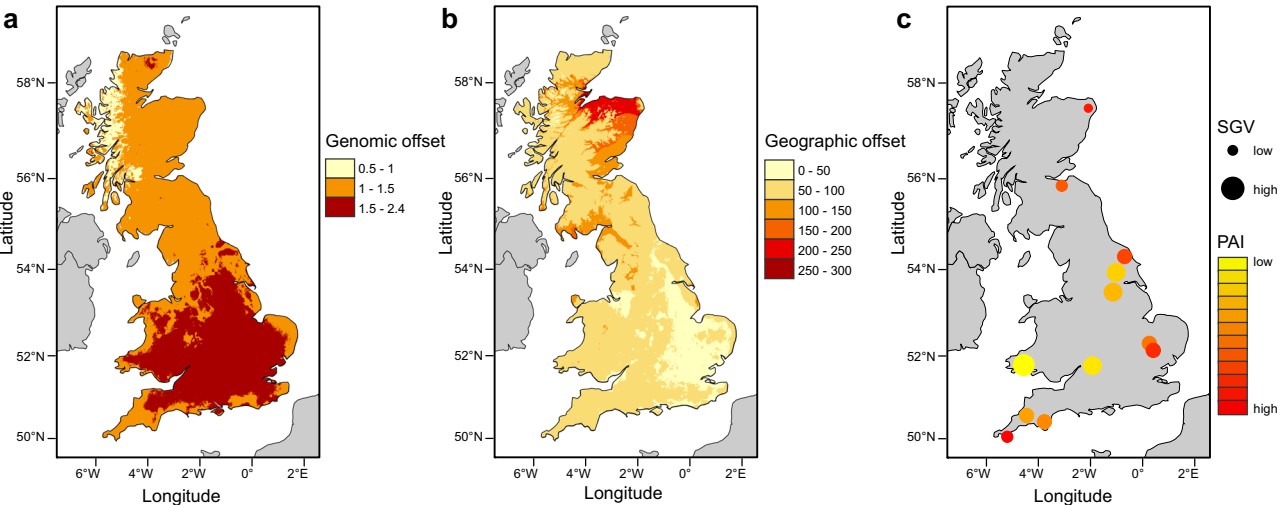

**Fig. 4 | Predicted disruption of local adaptation by climate change and the potential for future adaptation. a** Predicted genomic offset; and **b** geographic offset for the bank vole in Britain for 2081–2100; and **c** adaptive standing variation (SGV) and population adaptive index (PAI). Spatial mapping of genomic offset identified the populations in central England as being most vulnerable to future climate change. In contrast, the geographic offset showed that populations in Scotland are physically furthest from those with relevant adaptive variation (eastern England). The maps were created using ArcMap (v. 10.8) and the Esri World Countries dataset (https://www.esri.com).

alleles between populations may therefore be a viable adaptation mechanism in this system.

These results contribute to a recent body of evidence that implicates multilocus climatic adaptation across many species[8,9,42–44]. This aligns with expectations that many climate-associated traits are genetically complex, with variation among populations arising from small contributions at many segregating sites[31]. Polygenic traits may respond quickly to environmental change because the time to a new adaptive optimum is inversely proportional to the number of loci controlling the trait[45]. Here, we expand the set of candidate genes for bank voles to include 126 SNPs throughout the genome involved in a range of processes (Fig. 2). Gene ontology indicates enrichment associated with the regulation of cellular, biological and developmental processes, many of which link to heat or oxidative stress. Heat stress can increase cellular energy demands, driving the overproduction of reactive oxygen species (ROS)[46] and inducing oxidative stress[47–49]. Organisms respond to this stress through multiple mechanisms that prevent nucleic acid, protein and cellular damage[40,50,51]. Among bank voles, we have detected adaptive candidate genes (e.g., *Adprs*, *Dhcr24*, *Epas1*, *Prkca, Prkn,* and *Txn2*) implicated in responses to oxidative stress and reduction of oxidative damage. For example, *Prkca* is essential in the regulation of oxidative stress, shaping cellular processes by activating the *mTOR* signaling pathway with impacts on actin stress fibers[52,53]. An additional set of candidate genes (e.g., *Hcls1*, *Prkn, Sptbn5* and *Tln1*) encode actin-binding proteins, regulate dynamics (i.e., assembly and disassembly) of actin filaments, and reorganize F-actin proteins (e.g., *Arhgap32*) during oxidative stress[54–57]. In particular, *Arhgap32* was one of two genes with two nonsynonymous mutations and had the same allele fixed, or almost fixed, on both sides of the temperature gradient (Supplementary Fig. 3), indicating it may mediate stress response at thermal extremes[49,57].

Unexpectedly, ~16% of adaptive genes identified in British bank voles have previously been linked to responses to hypoxia in mammals in high-altitude environments[9,58] (Supplementary Data 1). Two genes (*HBB-T1*, *Epas1*) are among the top candidates for hypoxia adaptation at high elevation in deer mice[9]. As little altitudinal variation was represented in our study (from 0 to 207 m, an elevation with 98% of the oxygen present at sea level), selective pressures on these genes are unlikely to be linked to altitude. Instead, these genes were associated with the annual mean temperature (BIO 01), suggesting

selection of *HBB-T1* and *Epas1* may also be important in adaptation to thermal and heat-induced oxidative stress. Heat stress causes excessive oxygen demand in cells[50] that can lead to inadequate oxygen delivery to the tissues. Consistent with this phenomenon, we detected several genes related to hypoxia (e.g., *F7*, *mTOR*, *Prkca* and *Tnx2*) that may be important in the regulation of oxygen homeostasis and blood flow via contraction of the vascular smooth muscle and vasoconstriction under hypoxia[59,60]. Similar modifications of adaptive pulmonary function have been described in high-altitude deer mice[9]. Biological and cellular adaptations to heat may also improve tolerance to hypoxic or ischemic stress, as well as cross-tolerance to different stress stimuli[61]. These and other genes identified herein support a pattern of cross-adaptation along thermal and hypoxic gradients (similar to refs. 62,63), with the same genes responding to low oxygen at high altitudes and heat-induced hypoxia at low altitudes. Variation in these genes may be important in adaptation to climate change, particularly in populations and species where elevational range shifts are likely.

As global warming increases temperature and drives oxidative stress[64], understanding the geographic distribution of this variation and the underlying mechanisms (e.g., antioxidative defense) and drivers (e.g., thermal and hypoxic cross-adaptation) becomes increasingly important. Our findings support a variety of adaptive mechanisms against heat and oxidative stress[65–67] resulting from the cumulative effects of multiple adaptive loci of small effect which vary along environmental gradients. The characteristics and current distribution of standing adaptive variation suggest that bank voles throughout Britain can maintain climate adaptation under warming predicted for the next few decades, but an influx of adaptive variation from southern populations to those along the Celtic fringe may be important in preventing local maladaptation. The degree to which this pattern of relatively high peripheral vulnerability and central resilience is widespread, and the potential for genomic composition to shift in response to climate change, is key to the fate of biodiversity in a future, warmer world.

## Methods
### DNA extraction and sequencing
DNA was isolated using the DNeasy kit (Qiagen, Valencia, CA) from tissue samples (liver, spleen, and toes) originating from 12 sites across

Britain (Fig. 1b; Supplementary Table 1). The quality and quantity of extracted DNA were assessed by fluorometry (Qubit fluorometer, Life Technologies, Carlsbad, CA) and gel electrophoresis. Short-read genomic libraries were constructed using the NEBNext Ultra II FS DNA Library Prep Kit (New England Laboratories, Woburn, MA) according to the manufacturer's protocol. Extracted DNA for each vole was fragmented using enzymatic fragmentation. NEBNext adapters were ligated to fragments from each reaction, amplified by five cycles of PCR by using index primers (NEBNext Multiplex Oligos) and purified using NEBNext Sample Purification beads. To verify the appropriate quantity of incorporated adapters and the correct fragment size, an electrophoresis TapeStation (Agilent, Santa Clara CA) was used. Finally, 150 bp paired-end sequencing on the Illumina NovaSeq 6000 sequencing platform was performed at the Oklahoma Medical Research Foundation DNA Sequencing Facility (University of Oklahoma, OK). Raw sequencing data have been deposited in the NCBI Sequence Read Archive under BioProject accession number PRJNA1017835.

### Read mapping and SNP calling

Sequencing yielded an average of $1.3 \times 10^8$ reads per vole (range $6.2 \times 10^7 - 2.2 \times 10^8$). Data were demultiplexed and converted to fastq formats with bcl2fastq conversion software (Illumina, v. 2.20.0.422). Quality control checks were performed by FastQC (v. 0.11.9)[68,69]. Reads were then processed with trimmomatic (v. 0.36)[70] to remove adapters (ILLUMINACLIP 2:30:10:8) and low-quality bases (SLIDINGWINDOW:4:15; MINLEN:30). After filtering, we verified efficient adapter trimming (i.e., no samples remaining with adapter contamination >0.1%). Reads for each individual were then mapped to the reference genome[71,72] using bwa-mem algorithm of bwa (v. 0.7.10-r789)[73] and resulting files were sorted and indexed with samtools (v. 1.6.0)[74]. Duplicate reads were identified with picardtools (v. 2.18.5-6) MarkDuplicates (http://broadinstitute.github.io/picard) and removed. Reads mapped well to the reference genome (only <1% pairs were not mapped). The mean coverage across all individuals was 6× (range 3×–46×). Prior to SNP calling, we down-sampled reads from two individuals sequenced with the highest coverage (44× and 46×) to approximately 15% of reads. SNP calling was then performed using samtools mpileup (-q 30, -Q 20, -t DP, ADF, ADR, PL options). Afterward, bcftools (v. 1.13)[74] call was executed to call SNPs at each genomic position that passed through the quality control of the samtools mpileup step. We restricted our analysis to the 82 longest scaffolds in the reference genome, which account for over 99% of the genome length and include 28 chromosome-level scaffolds, excluding the scaffold representing the X chromosome. We then performed initial filtering with bcftools, removing SNPs that did not meet basic quality requirements (-g 5, -e DP < 365 and DP > 1097, Q < 30, MQ < 30, MQ0F > 0.1), and vcftools (v. 0.1.17) was used to remove indels and SNPs with more than 50% missing genotypes. After this initial filtering step, the dataset contained 50,142,414 SNPs for 111 bank voles from Britain.

Next, we removed SNPs inside transposable elements (SINE, LINE, and LTR), tandem repeats (as defined by RepeatMasker v. 4.1.2 using cross-match algorithm and the RepBase rodent library[75–77]) and inside the areas with mappability scores less than 1 (as defined by GenMap v. 1.3.0 using -k 100 and -e 2 options[78]). To produce the final dataset, SNPs were quality-filtered, retaining those that satisfied the following criteria: bialellic SNPs (min-allele 2, max-allele 2), $p$-value below $10^{-6}$, mean depth above 5 (MEAN (FMT/DP) < 5), minor allele frequency (MAF) above 5%, fraction of missing genotypes less than 10% (F_MISSING > 0.1). To remove SNPs exhibiting departure from HWE, we used the best practice recommendation for filtering genomic data (see ref. 79), where Out Within (SNPs removed from sampling location where they are out of HWE) and HWE < 0.001 criteria were applied. Finally, we used plink v. 1.9 (the –indep-pairwise 50 5 0.5 flag[80]) to

generate a pruned subset of SNPs that are in approximate linkage equilibrium. After all filtering, the complete dataset contained 241,099 SNPs for 111 individuals from 12 localities across the bank vole distribution in Britain[72].

### Description of the population structure

We calculated pairwise $Fst$ values between each of the 12 British populations with EIGENSOFT v. 7.2.1[81] and then used the same software to perform Principal Component Analysis (PCA) as a two-dimensional summary of the observed genetic variation. We then used Admixture (v. 1.3.0)[82] to estimate the number of genomic clusters ($K$) and the proportions of the genome derived from these clusters for each vole. The value of parameter $K$ was estimated using the Admixture cross-validation procedure, which was run 10 times with random seeds, each time for values from 1 through 10, with 10 replications for each $K$.

### Identification of loci involved in local adaptation

We used a combination of three population differentiation and GEA methods to detect signatures of divergent selection in bank vole populations in Britain[37]. We classified SNPs identified as outliers by all three methods as three-way candidates for adaptive loci. Because these approaches use different methods to estimate genome-wide signatures of local adaptation and to correct for higher false positive rates associated with some approaches under certain scenarios (e.g., ref. 83), SNPs identified by all three approaches represent high-confidence adaptive targets given the demographic background.

The first two methods—the $F_{ST}$ outlier approach and pcadapt—identify SNPs that exhibit above-average differentiation among populations based on the assumption that alleles involved in local adaptation should occur at higher frequencies where they increase fitness and at lower frequencies where they decrease fitness[37]. We calculated $F_{ST}$[84] for each SNP within the set of populations (--within flag) in plink and considered the SNPs with the highest $F_{ST}$ values above the 95th percentile threshold as candidate SNPs involved in local adaptation. We then used the R package pcadapt (v. 4.3.3)[85] to identify SNPs whose allele frequency is excessively associated with the population structure characterized by PCA. The method performs correlative modeling between individual allele counts and the first $K$ PCs, quantifies the strength of such association by the robust Mahalanobis distance, and identifies outliers by converting Mahalanobis distances to $P$-values based on a chi-squared distribution with $K$ degrees of freedom. Catell's rule was applied to the scree plot of variance explained by the first 20 PCs to determine $K$ that best represents the observed population structure. Candidate loci were determined using a significance level equivalent to a genome-wide false discovery rate (FDR) of 0.1 using the qvalue R package (v. 2.15.0)[86].

The third method was partial redundancy analysis (pRDA), a GEA approach that identifies SNPs whose allele frequencies show an unusually strong correlation with environmental variables, suggesting that the alleles provide a selective advantage in a particular environment[37]. pRDA uses constrained multivariate ordination to model linear relationships between genetic variation and environmental variables that are summarized by PCA[33]. There are two main potentially confounding factors in GEA analysis: population structure driven by genetic drift with reduced gene flow, which can result in genetic patterns similar to those caused by local adaptation, and recent demographic expansion, which can generate gradients in allele frequencies that are correlated with environmental variables that happen to change along the direction of expansion (e.g., north-south), thus mimicking a pattern of selective sweep or local adaptation[29,32]. To account for such potential confounding effects, pRDA adjusts the relationship between genomic response and environmental predictor variables through a set of conditioning variables, allowing partitioning of the percentage of genetic variance explained by each factor[32,33,39]. pRDA is capable of identifying signatures of weak and/or multilocus selection and

adaptation[35,87], and has been shown to have a low rate of false positives and a high rate of true positives across a range of demographic histories[87] (but see ref. 83).

We obtained bioclimatic variables from WorldClim v.2[88] at 30-arc-second resolution and used the pairs.panels function of the vegan (v. 2.6.4) package[89] in R to select a set of uncorrelated variables based on thresholds r < 0.7 and variance inflation factor < $10^{87,90}$. We used population ancestry coefficients from Admixture analyses for $K = 3$ and geographic coordinates of populations as two sets of conditioning variables to estimate the contribution of population structure and geography, respectively, to the observed pattern of genetic variation. We performed pRDA-based partitioning of the percentage of genetic variance explained by each specific set of predictor or conditioning variables using the rda function of the vegan package.

To identify candidate loci associated with environmental predictors, we performed a pRDA in vegan[33] with the first three RDA axes explaining most of the genetic variance associated with the predictors and conditioned the analysis using ancestry coefficients as proxies for neutral genetic differentiation due to isolation-by-distance and demographic history, such as end-glacial colonization[32]. Similar to pcadapt, we controlled for an FDR by converting P-values to q-values using the qvalue R package. Two different FDR values were used by pRDA to balance the known potential for a high proportion of false positives that may be associated with these approaches[83] with the desire to minimize false negatives (i.e., true outlier loci eliminated by being overly stringent). Therefore, for most analyses, we focused on SNPs identified as outliers by all three methods but balanced this stringency by allowing a higher FDR (0.1; equivalent to that recommended for pcadapt). We then applied the more stringent cutoff (0.05) for the FDR to identify the adaptively enriched genetic space for our predictions of the genomic offset[32,38].

## Modeling adaptation gradients across space and time

To understand the impact of adaptive variation across the landscape and under future climates, we subjected the set of SNPs representing the adaptively enriched genetic space to a second pRDA[32,38]. We first used the scores of the environmental predictors on the RDA axes to calculate the adaptive index of individuals in the current environment[32,33]. The adaptive index was estimated for each of the three RDA axes and each environmental pixel. We then extrapolated the genetic-environment relationship determined by pRDA to future environments to predict a potential shift in the adaptive optimum due to future climate change. We applied the same procedure used for the current climate to predict the optimal adaptive index using future environmental conditions represented by an average of projections for each of the four bioclimatic variables from six general circulation models for 2081–2100: ACCESS-ESM1.5[91], CNRM-CM6-1-HR[92], FIO-ESM-2-0[93], GISS-E2-1-G, GISS-E2-1-H[94], and MIROC6[95]. In each model, we used the scenario with the highest greenhouse gas emissions (SSP5-8.5), which assumes continued growth of the fossil fuel-based global economy and projects the largest temperature increase[96]. The Euclidean distance between the adaptive index under current and future environmental conditions for each pixel provided an estimate of the genetic offset, i.e., the shift in adaptive genetic composition in a population that would be required to track predicted climate change. We summed the genetic offset across the three RDA axes and plotted it on a map to show the magnitude of genomic mismatch (i.e., maladaptation) for a population under future climate change. We used ArcMap (v. 10.8) and the Esri World Countries dataset (www.esri.com) to create the maps.

We then measured the geographic distance for each pixel in the future adaptive landscape to the nearest population that has an equivalent adaptive index under current climatic conditions. This geographic offset estimates the potential for alleles favorable under future climatic conditions to arrive by migration from another population[32].

Finally, we quantified the amount of adaptive standing genetic variation (SGV) in each population by calculating the mean of the variances of allele frequencies across the adaptive candidate loci[32,97]. For comparison, we also calculated SGV for putative neutral SNPs, i.e., SNPs that were not identified as outliers by any method. We then estimated the population adaptive index (PAI), a measure of the difference between the adaptive allele frequencies of a given population and the mean adaptive allele frequencies of all populations[98]. This provides insights into how well-provisioned a population is to adapt in the future (e.g., ref. 32), with higher values indicating populations where most adaptive loci are fixed, or nearly fixed, and further response to selection is less likely.

For comparison, we also estimated genomic and geographic offsets with GF using the approach of Fitzpatrick and Keller[34,99]. Similar to pRDA, GF predicts genomic offset as the differentiation for climatically adaptive SNPs between present and future climates at the same location and a geographic offset as the distance from a location in the future climate to a location with minimal offset in the current climate[34,99]. We used the R package gradientforest (v. 0.1.34)[100] and fitted GF with 2000 regression trees per SNP, a variable correlation threshold of 0.5, and default values for the other parameters[34].

## Functional annotation and ontology of adaptive candidate loci
To understand the potential impacts of the three-way candidate loci (i.e., SNPs those identified as outliers by $F_{ST}$, pcadapt, and pRDA) on functional changes in bank voles, we used the CLC Genomics Workbench (v. 23.0.2; QIAGEN, Aarhus, Denmark) and the positions of each SNP along the bank vole reference genome to identify SNPs located within coding regions and determined whether changes were synonymous or nonsynonymous. For SNPs located within or less than 1000 bp upstream or downstream of known genes in the bank vole genome, we identified the orthologous *Mus musculus* gene using the UniProt database[101] to find the functional annotation of the gene and search for its involvement in adaptation in other species. To determine whether the list of annotated candidate genes was overrepresented for specific molecular functions, biological processes and cell components[102], we performed Gene Ontology (GO)-term enrichment analysis using Panther v. 17.0[103] and an FDR adjusted P-value (q-value) of 0.05. In addition, we used QuickGO v. 2022-11-18[104] to identify the involvement of the genes in particular sub-processes, e.g., response to oxidative stress (GO:0006979), response to hypoxia (GO:0001666) and actin binding (GO:0003779). The Manhattan plots and distribution of allele frequencies were generated using the ggplot2 R package (v. 3.4.2)[105].

## Reporting summary
Further information on research design is available in the Nature Portfolio Reporting Summary linked to this article.

## Data availability
The raw Illumina sequencing data generated in this study are deposited at NCBI in the Sequence Read Archive (SRA) under BioProject PRJNA1017835 with accession numbers SRR26070875-SRR26070988. SNP genotype data, reference genome assembly and annotation files are available in the Dryad Digital Repository (https://doi.org/10.5061/dryad.kwh70rz96). WorldClim climate data are publicly available at (https://www.worldclim.org).

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

## Acknowledgements

This study was supported by the Czech Science Foundation (grant no. 20-11058S, P.K., S.M., M.H. and M.A.E.) and Ministry of Education, Youth and Sports of the Czech Republic (grant no. EXCELLENCE CZ.02.1.01/0.0/0.0/15_003/0000460 OP RDE, M.A.E. and P.K.). Part of the computations for this study were performed at the OU Regional Supercomputing Center for Education and Research (OSCER) and the Poznan Supercomputing and Networking Center.

## Author contributions

P.K., S.M., H.C.L. and J.B.S. designed the study. S.M., M.A.E., H.C.L., M.O.R.d.C., M.K. and M.H. performed the data analysis with assistance from P.K. All authors contributed to interpreting the results. S.M., H.C.L. and P.K. led the writing of the manuscript with input from J.B.S., L.J.W., M.A.E., M.O.K.d.C., M.K. and M.H.

## Competing interests

The authors declare no competing interests.
