## [Peer Review File · Nature Communications]

Local adaptation and future climate vulnerability in a wild rodentREVIEWER COMMENTS

Reviewer #1 (Remarks to the Author):

I thoroughly enjoyed reviewing this paper, as it sheds light on an important topic in climate change research. Marková et al. analyzed the effects of local adaptation and future anthropogenic climatic change of a bank vole in Britain. The authors identified population structure that was likely due to a two-phase colonization process after the LGM. They also determined SNPs under selection, using three different methods, and showed the adaptive landscape across geography for present and future conditions. The authors concluded that climate has been a major driver of local adaptive variation and the importance of considering local conditions when determining how diversity is spread across the landscape. This manuscript is concise, well written, and provides an important contribution to the field. However, clarification on several key methodological points is needed. Without this clarification of the analyses the paper is lacking. The points below could be considered to increase the impact of the paper.

Comments:

Lines 87-88: The HbS and HbF notation is confusing and I'm not sure it provides the reader with essential information.

Lines 110: I suggest the authors change from "By sequencing 111 complete genomes" to something that indicates whole genome resequencing. I think this more accurately reflects what the authors did here. Also, the cited article (which the lead author here is also a lead author on) says the bank vole genome project "obtained an assembly almost at the chromosome level". This distinction is important because I was confused in the methods portion where the authors are discussing scaffolds used in this study. Also, is the genome freely available and could a link be provided?

Lines 110-120: I think it's important for the authors to mention they are only looking at a subset of the range for this species.

Line 128: I would remove the word "optimal" and replace with "best fit model"

Lines 150-152: Why do the authors use two different FDR values for pRDA in this manuscript? If the authors use a 0.05 FDR for the initial pRDA analysis, how many overlapping SNPs would have been discovered? I know they mention in the methods they implemented "a more stringent criterion", but I'm curious about why they choose these values.

Lines 158-161: The authors state that northwestern and southeastern populations live in distinct environmental conditions; however, they have no samples from these regions. Do the authors think this pattern seen in Fig. 2A could be due to how they extrapolated the RDA over the landscape? Are 12 sampling sites enough for large landscape analyses?

Line 165: I think it would be good for the authors to compare their results with Fitzpatrick and Keller (2015) as a proof of concept (either GDM or GF). They have the data, pairwise F_{st} and climate, to easily do this. I appreciate the authors developing their own methods, but it would be good for the reader to see how it compares.

Fitzpatrick, M.C., and Keller, S.R. 2015. Ecological genomics meets community-level modelling of biodiversity: Mapping the genomic landscape of current and future environmental adaptation. *Ecology letters* 18 (1): 1-16.

Line 197: Comma after the word "Here".

Line 456: Change "Fig. 1" to "Fig. 1b".

Lines 465-467: Could the authors elaborate on the sequencing depth they achieved in this study?

Line 485: Did the authors assess the effect of missing data on downstream analyses? 50% threshold for missing data seems high.

Line 486: Is the number "50" a typo or did the authors get over 50 million SNPs? I also think the authors can remove the word "genotype".

Lines 468-487: Some of the software indicted in the text is not cited here, but it appears in the references. Additionally, this section was very confusing for me as a reader. The authors discuss SNPs filtering first, then scaffolding, and finish with more SNP filtering. I think it makes more sense to talk about the scaffolding first and then discuss all the SNP filtering done by the authors.

Lines 488-500: Are the authors using the words "SNP" and "loci" interchangeably here? This paragraph is confusing because I think of SNPs and loci and different concepts (e.g., there can be multiple SNPs per locus). Are the authors using one biallelic SNP per locus (if there are 241,099 loci are there 241,099 SNPs)?

Line 502: What program was used to calculate pairwise Fst?

Line 504: How did the authors determine the best fit model for K?

Line 543: What resolution of the wordclim layers were used?

Line 583: The authors calculated SGV for adapted loci, it would be interesting to see the SGV across loci they determined were effectively neutral.

Discussion: I think this manuscript would benefit from the authors describing the relationship of genomic vulnerability with population density as well. The authors state "Our models predict increased climate vulnerability (i.e., high genomic offset) for populations in central England, but these populations may not face great risks of extirpation because they are relatively close to sources of suitable adaptive alleles (i.e., a small geographic offset)."; however, that genomic vulnerability is around London and the surrounding areas. Although its geographically close to other populations maybe the landscape won't allow for the exchange of necessary alleles.

Fig. 3: Should it be changed to "SGV"

Fig. 4: Why is this figure not referenced earlier in the manuscript? I think it should be referenced in the "Genomic signatures of local adaptation" section. This will change the Figure order.

SFig. 1 appears blurry. I'm not sure fig. b adds much to this manuscript.

SFig. 3: This figure would benefit from a map (it doesn't have to be very large) so the reader doesn't have to go back and forth to understand the geographic patterns. Also, the authors should state what the two colors indicate.

Reviewer #2 (Remarks to the Author):

Marková and collaborators evaluated genetic-environment associations and identified potential adaptive variation, based on whole-genome sequencing, of the bank vole *Clethrionomys glareolus* from to regions (Scotland and England and Wales). There is a rich source of information for this species from studies performed by some of the coauthors of the present manuscript, including their colonization routes and processes and its polymorphism patterns in haemoglobin (Hb), for which they describe the potential role of such variation and local adaptation of the bank voles under future climate change.

All this background and the original questions and sound methodology performed in the present study, allowed them to accomplish an extraordinary research. Based on 111 sequenced genomes

aligned to a chromosome-level assembly, they were able to identify outlier loci and functional genes, associated with environmental factors and, potentially, key for local adaptation. They also performed future climate models to assess the availability of standing variation for the populations' future adaptation. Their findings reveal that the cumulative effect of different adaptive loci is associated with some adaptive mechanisms related with oxidative stress. Also, and concordant with what they had found for the haemoglobin (HbS and HbF), these results indicate differences between the two regions, where the southern Britain population likely has a better chance of surviving global warming with its standing variation.

It is a well written, scientifically soundly supported study. The conclusions are original and of relevance for the genetics, evolutionary and mammalogy research. I congratulate the authors and have only two minor suggestions:

Line 26: add "an", so it reads "... warming in an iconic ..."

Lines: 197-199: "Population structure was strongly shaped by the two-phase colonization process, with isolation-by-distance and recent demographic expansion shaping the distribution of variation" On what is the statement of a recent demographic expansion based? The PCA and the admixture analyses and results, separating the two most geographically and genetic distant populations, does suggest a two-phase colonization, but to determine recent demographic expansion authors should have done specific analysis (e.g. skyline plots or other). Please clarify.

Reviewer #3 (Remarks to the Author):

This study explores patterns of local adaptation to climate in the bank vole & uses this information to assess future climate risks in the context of the potential for maladaptation and the need for an influx of climate-adaptive genes from other populations. I focused my review mainly on the statistical analyses and have only a few concerns.

I am not an expert in the application of (p)RDA as used in this study, but I am aware of a recent study Lotterhos (2023) PNAS that raised some substantial concerns regarding the use of this method for outlier detection & in particular the high rate of false positives. It would be good to see a response addressing whether this is an issue for the present study - in terms of both outlier detection and the application of (p)RDA for projecting genetic offsets.

My second major concern is in regards to how the magnitude of genetic offsets is discussed and presented in the context of risk / population extirpation. For example on L285-288, statements are made regarding the the ability of bank voles to survive near-term climate changes and the possibility of extirpation without influx of adaptive genes from other populations. However, the study does not link offsets to fitness / risk of extirpation so these statements are pure speculation. How do we know what level of offset projected from (p)RDA is tolerable / not tolerable? At best, the only statements that can be made is that some populations have greater / lower offsets than others and therefore are at higher / lower risk - but in no way can the magnitude of offset be directly linked to the magnitude of risk. It would be better to use terms like "larger" / "smaller" rather than "high / low".

Suggest using the term "climate change" instead of "global warming" throughout the MS.

Response to Reviewers

Reviewers' comments are in plain text. Authors' responses are in bold. The line numbers in the responses correspond to those in the revised manuscript file, in which the changes are indicated by highlighting.

Reviewer #1 (Remarks to the Author):

I thoroughly enjoyed reviewing this paper, as it sheds light on an important topic in climate change research. Marková et al. analyzed the effects of local adaptation and future anthropogenic climatic change of a bank vole in Britain. The authors identified population structure that was likely due to a two-phase colonization process after the LGM. They also determined SNPs under selection, using three different methods, and showed the adaptive landscape across geography for present and future conditions. The authors concluded that climate has been a major driver of local adaptive variation and the importance of considering local conditions when determining how diversity is spread across the landscape. This manuscript is concise, well written, and provides an important contribution to the field. However, clarification on several key methodological points is needed. Without this clarification of the analyses the paper is lacking. The points below could be considered to increase the impact of the paper.

We thank the reviewer for the positive comments.

Comments:

Lines 87-88: The HbS and HbF notation is confusing and I'm not sure it provides the reader with essential information.

As suggested, we have removed the HbS and HbF notations, which were originally included as a link to previous studies of bank vole haemoglobin.

Lines 110: I suggest the authors change from "By sequencing 111 complete genomes" to something that indicates whole genome resequencing. I think this more accurately reflects what the authors did here. Also, the cited article (which the lead author here is also a lead author on) says the bank vole genome project "obtained an assembly almost at the chromosome level". This distinction is important because I was confused in the methods portion where the authors are discussing scaffolds used in this study. Also, is the genome freely available and could a link be provided?

We have rephrased the sentence to indicate that the approach we are using is "whole genome resequencing".

By "an assembly almost at the chromosome level", we mean an assembly in which a single large scaffold corresponds to a chromosome (two large scaffolds in the case of one

chromosome). In addition, there are several smaller scaffolds that cannot currently be assigned to a chromosome but may contain adaptive genes and were therefore included in the analysis.

We have changed the wording of the relevant portion of the Methods (lines 348-353) to clarify that we restricted our analysis to the 82 longest scaffolds in the reference genome, which account for over 99% of the genome length and include 28 chromosome-level scaffolds, excluding the scaffold representing the X chromosome.

The bank vole reference genome assembly was deposited at NCBI under BioProject ID PRJNA1005562. We have included the BioProject ID in the Data availability statement. The BioProject will be publicly available once a detailed report of the genome is published.

Lines 110-120: I think it's important for the authors to mention they are only looking at a subset of the range for this species.

We have added a sentence that justifies the focus on Britain.

Line 128: I would remove the word "optimal" and replace with "best fit model"

We changed "optimal for this dataset" to "the best fit for this dataset".

Lines 150-152: Why do the authors use two different FDR values for pRDA in this manuscript? If the authors use a 0.05 FDR for the initial pRDA analysis, how many overlapping SNPs would have been discovered? I know they mention in the methods they implemented "a more stringent criterion", but I'm curious about why they choose these values.

Two different FDR values were used by the pRDA to try and balance the known potential for a high proportion of false positives associated with these approaches with the desire to minimize false negatives (i.e., true outlier loci that were eliminated by being overly stringent). Here, we did that by focusing the bulk of analyses on SNPs identified as outliers by all three methods, but we balanced that stringency by allowing a higher FDR (equivalent to that recommended for padapt). Then we applied the more stringent cutoff for the FDR in order to identify the "adaptively enriched genetic space" for our predictions of genomic offset.

By initially applying a higher FDR for the pRDA we were able to consider a larger number of overlapping candidate loci (1075); if we had used an initial 0.05 FDR our candidate set would have consisted of 741 overlapping loci.

We have explained the reasons for using two different FDR values in the Methods section (lines 429-436).

Lines 158-161: The authors state that northwestern and southeastern populations live in distinct environmental conditions; however, they have no samples from these regions. Do the

authors think this pattern seen in Fig. 2A could be due to how they extrapolated the RDA over the landscape? Are 12 sampling sites enough for large landscape analyses?

We recognize that the southeast of Britain is represented by two sites (CAM1 and CAM2), while the conclusion for the northwest of Scotland is based on the extrapolation of the adaptive index to the climatic data, following the approach of Capblancq et al. (2020). During the project development we had hoped to have rather better geographical coverage but were unable to conduct fieldwork (or obtain suitable museum samples) due to Covid shutdowns.

In response to the reviewer's comment, we have acknowledged in the manuscript that northwest Scotland is estimated to be at the extreme end of the gradient of adaptation to cold and wet climates (lowest RDA1 index), but this should be interpreted with caution because we did not sample this area and the extrapolation may not accurately reflect genomic differentiation in this region (lines 166-169).

However, even if the predicted adaptive index for northwest Scotland were not entirely accurate, this would not change the main pattern of inferred adaptive turnover between the northwest (generally Scotland and Wales) and the southeast of Britain (essentially England), which reflects the expected gradient of climatically driven selection pressure in Britain, manifested as a transition from a relatively wet and cool northwest fringe to a drier and warmer area near continental Europe.

Therefore, although the sampling design in the present study represents a trade-off between the number of voles and the number of localities, and is limited by the availability of suitable samples from some parts of Britain, it is unlikely that the main patterns inferred are an artifact of the extrapolation.

Line 165: I think it would be good for the authors to compare their results with Fitzpatrick and Keller (2015) as a proof of concept (either GDM or GF). They have the data, pairwise F_{st} and climate, to easily do this. I appreciate the authors developing their own methods, but it would be good for the reader to see how it compares.

Fitzpatrick, M.C., and Keller, S.R. 2015. Ecological genomics meets community-level modelling of biodiversity: Mapping the genomic landscape of current and future environmental adaptation. *Ecology letters* 18 (1): 1-16.

At the recommendation of the reviewer, we applied the gradient forest (GF) method (Fitzpatrick and Keller, 2015; Gougherty, Keller & Fitzpatrick, 2021) to predict the genomic and geographic offset. The results of the GF method are similar to those of the RDA approach, confirming that populations in England are likely to acquire the adaptive variation they need to adapt to future global warming more readily (lowest geographic offset) than other parts of Britain, although they will have the greatest need for adaptive genetic change (highest genomic offset).

We have included the GF analysis in the manuscript (lines 177-180, 471-477) and show the GF results in Supplementary Fig. 4.

Line 197: Comma after the word “Here”.

We have made this change.

Line 456: Change “Fig. 1” to “Fig. 1b”.

We have made this change.

Lines 465-467: Could the authors elaborate on the sequencing depth they achieved in this study?

We added this information. Sequencing yielded an average of 1.3×10^8 reads per vole (range $6.2 \times 10^7 - 2.2 \times 10^8$), and mean coverage across all individuals was 6× (range 3× – 46×). Prior to SNP calling, we down-sampled reads from two individuals sequenced with the highest coverage (44× and 46×) to approximately 15% of reads (lines 333, 343-345).

Line 485: Did the authors assess the effect of missing data on downstream analyses? 50% threshold for missing data seems high.

We agree that a final missing genotype rate of 50% would be quite high and that its impact on the outcome should be considered. However, in the present study, we only considered a much lower rate of missing genotypes of 10%, which is commonly used in population genomics studies. We revised the Methods to clarify that a criterion of 50% missing genotypes was only used in the initial filtering after SNP calling to remove SNPs that did not meet basic quality requirements, but for the final dataset a missing genotype rate of less than 10% was applied.

Line 486: Is the number “50” a typo or did the authors get over 50 million SNPs? I also think the authors can remove the word “genotype”.

We removed “genotypes” and corrected the missing comma in the total number of 50,142,414 SNPs that were called (before the filtering that yielded the final dataset).

Lines 468-487: Some of the software indicted in the text is not cited here, but it appears in the references. Additionally, this section was very confusing for me as a reader. The authors discuss SNPs filtering first, then scaffolding, and finish with more SNP filtering. I think it makes more sense to talk about the scaffolding first and then discuss all the SNP filtering done by the authors.

We have added the missing in-text citations and references are up-to date.

We have revised this section to clarify that we performed initial basic SNP filtering following SNP calling and then generated the final SNP dataset using more stringent filtering criteria. We did not perform scaffolding in this study because we used an already available reference genome assembly. The reference to scaffolds is to explain that we restricted the analysis to the 82 longest scaffolds in the reference genome, which accounted for over 99% of the genome length and included 28 chromosome-level scaffolds, excluding the X chromosome. We have clarified this in the text.

Lines 488-500: Are the authors using the words “SNP” and “loci” interchangeably here? This paragraph is confusing because I think of SNPs and loci as different concepts (e.g., there can be multiple SNPs per locus). Are the authors using one biallelic SNP per locus (if there are 241,099 loci are there 241,099 SNPs)?

Given the nature of this dataset (i.e., variable sites from whole genome resequencing), SNPs are synonymous with loci (i.e., there are 241,099 SNPs, each at a different location in the genome). We have checked, and if necessary edited, the text throughout the manuscript to be consistent where we refer to SNPs (i.e., variable sites in the genome), versus loci (the variable site with an assumed function, e.g., candidate loci for adaptation), as well as where we are referring to genes (i.e., those SNPs/loci which are in close proximity to a region of the genome that encodes a function).

Line 502: What program was used to calculate pairwise Fst?

Both the pairwise Fst and PCA calculations were performed using EIGENSOFT. We have edited the Methods to more clearly describe our approach.

Line 504: How did the authors determine the best fit model for K?

The Methods now explain that the value of parameter K was estimated using the Admixture cross-validation procedure, which was run 10 times with random seeds, each time for values from 1 through 10, with 10 replications for each K.

Line 543: What resolution of the wordclim layers were used?

The WorldClim layers have a resolution of 30 arc-seconds. This is now included in the manuscript.

Line 583: The authors calculated SGV for adapted loci, it would be interesting to see the SGV across loci they determined were effectively neutral.

As suggested, we have now calculated SGV for putative neutral SNPs, i.e., SNPs that were not identified as outliers by any method. In this context, SGV for adaptive candidates may provide information on the adaptive variation available to each population as a substrate for selection under future climates. In contrast, SGV for neutral loci should have no such effect on adaptive

capacity. Interestingly, SGV at adaptive loci was slightly higher than at putative neutral loci for all populations (Supplementary Fig. 5), supporting the importance of SGV in local adaptation. This is now mentioned in the manuscript.

Discussion: I think this manuscript would benefit from the authors describing the relationship of genomic vulnerability with population density as well. The authors state “Our models predict increased climate vulnerability (i.e., high genomic offset) for populations in central England, but these populations may not face great risks of extirpation because they are relatively close to sources of suitable adaptive alleles (i.e., a small geographic offset).”; however, that genomic vulnerability is around London and the surrounding areas. Although its geographically close to other populations maybe the landscape won’t allow for the exchange of necessary alleles.

We agree that adaptation by alleles acquired from other populations depends on population density/size and habitat connectivity that facilitates gene flow. Although assessing these factors was beyond the scope of our study, we have now mentioned in the Discussion that in addition to the distance that alleles must travel, adaptation through gene flow also depends on population connectivity, and habitat fragmentation may prevent migration of adaptive alleles even if they occur nearby. The bank vole is common in brushy edge habitats in Britain, and its population structure is not particularly high for a small mammal, suggesting that there are no strong geographic barriers and that the spread of adaptive alleles between populations may be a viable adaptation mechanism (lines 257-262).

Fig. 3: Should it be changed to “SGV”

We have corrected this.

Fig. 4: Why is this figure not referenced earlier in the manuscript? I think it should be referenced in the “Genomic signatures of local adaptation” section. This will change the Figure order.

We have added a reference to this figure (now Fig. 2) in the section “Genomic signatures of local adaptation” and renumbered the other figures accordingly.

SFig. 1 appears blurry. I’m not sure fig. b adds much to this manuscript.

We have revised the figure to improve its clarity and removed panel b.

SFig. 3: This figure would benefit from a map (it doesn’t have to be very large) so the reader doesn’t have to go back and forth to understand the geographic patterns. Also, the authors should state what the two colors indicate.

As suggested, we have added a map to this figure, which shows the sampling sites, and we mentioned that the frequency of the alternative alleles is shown in blue and red.

Reviewer #2 (Remarks to the Author):

Marková and collaborators evaluated genetic-environment associations and identified potential adaptive variation, based on whole-genome sequencing, of the bank vole *Clethrionomys glareolus* from two regions (Scotland and England and Wales). There is a rich source of information for this species from studies performed by some of the coauthors of the present manuscript, including their colonization routes and processes and its polymorphism patterns in haemoglobin (Hb), for which they describe the potential role of such variation and local adaptation of the bank voles under future climate change.

All this background and the original questions and sound methodology performed in the present study, allowed them to accomplish an extraordinary research. Based on 111 sequenced genomes aligned to a chromosome-level assembly, they were able to identify outlier loci and functional genes, associated with environmental factors and, potentially, key for local adaptation. They also performed future climate models to assess the availability of standing variation for the populations' future adaptation. Their findings reveal that the cumulative effect of different adaptive loci is associated with some adaptive mechanisms related with oxidative stress. Also, and concordant with what they had found for the haemoglobin (HbS and HbF), these results indicate differences between the two regions, where the southern Britain population likely has a better chance of surviving global warming with its standing variation.

It is a well written, scientifically soundly supported study. The conclusions are original and of relevance for the genetics, evolutionary and mammalogy research. I congratulate the authors and have only two minor suggestions:

We thank the reviewer for the positive comments.

Line 26: add "an", so it reads "... warming in an iconic ..."

We have made this change.

Lines: 197-199: "Population structure was strongly shaped by the two-phase colonization process, with isolation-by-distance and recent demographic expansion shaping the distribution of variation"

On what is the statement of a recent demographic expansion based? The PCA and the admixture analyses and results, separating the two most geographically and genetic distant populations, does suggest a two-phase colonization, but to determine recent demographic expansion authors should have done specific analysis (e.g. skyline plots or other). Please clarify.

We agree that our results do not provide evidence of a recent demographic expansion. We have rephrased the sentence to clarify that the population structure was strongly shaped by the two-phase colonization process, with rapid population expansion and admixture after the Younger Dryas and recent isolation-by-distance shaping the distribution of variation (Fig. 1a,b).

Reviewer #3 (Remarks to the Author):

This study explores patterns of local adaptation to climate in the bank vole & uses this information to assess future climate risks in the context of the potential for maladaptation and the need for an influx of climate-adaptive genes from other populations. I focused my review mainly on the statistical analyses and have only a few concerns.

We thank the reviewer for taking the time to review our manuscript.

I am not an expert in the application of (p)RDA as used in this study, but I am aware of a recent study Lotterhos (2023) PNAS that raised some substantial concerns regarding the use of this method for outlier detection & in particular the high rate of false positives. It would be good to see a response addressing whether this is an issue for the present study - in terms of both outlier detection and the application of (p)RDA for projecting genetic offsets.

We agree that it is concerning that RDA had high false positive rates under the simulation settings used by Lotterhos (2023). Those results are in stark contrast to previous simulations (from Lotterhos and Whitlock 2014 but analyzed in Forester et al., 2018) in which RDA had a low rate of false positives and a high rate of true positives across a range of demographic histories and sampling designs, with the rate of false positives being the lowest among the methods compared (Forester et al., 2018).

To reduce the impacts of false positives on our analyses of genome wide loci and adaptive function, we considered SNPs as candidates for adaptive loci when identified by the intersection of three methods for population differentiation (pRDA, Fst, and padapt). By focusing on overlapping loci, false positive rates are known to be significantly reduced, albeit with some loss of true positives (e.g., Forester et al., 2018). Because a locus must be identified by all three approaches our results are unlikely to be substantially affected by the error of any single method (so, less susceptible to the false positive results identified by Lotterhos 2023). We have acknowledged both the concern and our approach to mitigating any biases that may arise from potential high false positive rates from the approach (lines 382-385, 415, 429-436).

To predict genomic offset, we used “adaptively enriched genetic space” as a multivariate response consisting of climate-related SNPs identified by RDA, using a more stringent false discovery rate to reduce the rate of false positives (see earlier reviewer response on this point). Furthermore, selection of genomic markers has been shown to have little effect on the predictive performance of genomic offsets that are not sensitive to the inclusion of non-adaptive genetic differentiation (Lachmuth, Capblancq, Keller & Fitzpatrick, 2023). Therefore, even if our adaptively enriched genetic space contained a high proportion of false positives, it would be unlikely to adversely impact our results.

My second major concern is in regards to how the magnitude of genetic offsets is discussed and presented in the context of risk / population extirpation. For example, on L285-288, statements are made regarding the ability of bank voles to survive near-term climate changes and the

possibility of extirpation without influx of adaptive genes from other populations. However, the study does not link offsets to fitness / risk of extirpation so these statements are pure speculation. How do we know what level of offset projected from (p)RDA is tolerable / not tolerable? At best, the only statements that can be made is that some populations have greater / lower offsets than others and therefore are at higher / lower risk - but in no way can the magnitude of offset be directly linked to the magnitude of risk. It would be better to use terms like "larger" / "smaller" rather than "high / low".

While genomic offsets are a promising tool for incorporating genomic information into climate change impact assessments, we agree that interpreting their absolute magnitude in a biologically meaningful way is challenging and in need of further study. We have reviewed and (where necessary) revised our language to ensure that comparisons are relative rather than absolute, in agreement with genomic offset studies by major players in the field of genomic offset estimation (e.g., Capblancq, Keller, Fitzpatrick). We were careful to use the terms “larger/smaller” or similar, rather than “high/low” when referring to genomic offset. We have rephrased the sentence in the last, generalizing paragraph of the Discussion to which the reviewer refers and in which we were originally more liberal, to suggest that an influx of adaptive variation from southern populations to those along the Celtic fringe may be important in preventing local maladaptation.

Although the genomic offset does not provide an estimate of the absolute risk of maladaptation, the pattern of adaptive turnover across Britain and the need for influx of adaptive variation from south to north is consistent with estimates from analyses requested by Reviewer 1 (see above). In addition, they align with those from a previous ENM study predicting disruption of the genetic-environmental association for the haemoglobin polymorphism in bank voles to such an extent that the north of Britain becomes unsuitable for the allele currently found there (Escalante et al., 2022).

Suggest using the term "climate change" instead of "global warming" throughout the MS.

At the reviewer’s suggestion, we have replaced the term "global warming" with "(future) climate change" throughout the manuscript, except in two instances, in the title and once in the last paragraph of the Discussion, where we have retained "global warming" as the predominant popular term. The use of the term "global warming" is consistent with the fact that the increase in global temperatures due to increasing concentrations of greenhouse gases is the main aspect and driving force of the projections of future climate models that we use.

REVIEWERS' COMMENTS

Reviewer #1 (Remarks to the Author):

I appreciate the hard work the authors have done to substantially improve their manuscript. I feel this is ready for publication.

Reviewer #3 (Remarks to the Author):

The authors have done a thorough job of addressing the majority of the concerns raised by myself and the other reviews. I feel the MS has improved as a result.

That said, I do feel like there are a few places that still require attention. In particular, in my previous review, I raised concerns regarding the language used to describe climate change risks as quantified using genomic / geographic offsets.

On L308-310, the authors state that "[t]he characteristics and current distribution of standing adaptive variation suggest that bank voles can survive throughout Britain under warming predicted for the next few decades." Not only does this statement conflict with statements made elsewhere in the Discussion (e.g., L245-251 that suggest certain populations "may not be able to compensate for climate change over the next few decades"), the study did not make any linkages between SGV and probability of survival. For that reason, I feel like these sorts of statements go beyond the nature of inferences that can rightly be concluded from the results.

It is important to keep in mind that these sorts of correlative modeling / forecasting exercises quantify aspects of relative exposure and the potential degree of relative maladaptation to climate change. In contrast, absolute vulnerability to climate change is multifaceted (see papers by Wendy Foden). Language should be used accordingly.

REVIEWERS' COMMENTS

Reviewer #1

I appreciate the hard work the authors have done to substantially improve their manuscript. I feel this is ready for publication.

We appreciate the positive comments.

Reviewer #3

The authors have done a thorough job of addressing the majority of the concerns raised by myself and the other reviews. I feel the MS has improved as a result.

We appreciate these positive comments.

That said, I do feel like there are a few places that still require attention. In particular, in my previous review, I raised concerns regarding the language used to describe climate change risks as quantified using genomic / geographic offsets.

On L308-310, the authors state that "[t]he characteristics and current distribution of standing adaptive variation suggest that bank voles can survive throughout Britain under warming predicted for the next few decades." Not only does this statement conflict with statements made elsewhere in the Discussion (e.g., L245-251 that suggest certain populations "may not be able to compensate for climate change over the next few decades"), the study did not make any linkages between SGV and probability of survival. For that reason, I feel like these sorts of statements go beyond the nature of inferences that can rightly be concluded from the results.

It is important to keep in mind that these sorts of correlative modeling / forecasting exercises quantify aspects of relative exposure and the potential degree of relative maladaptation to climate change. In contrast, absolute vulnerability to climate change is multifaceted (see papers by Wendy Foden). Language should be used accordingly.

We appreciate the reviewer's detailed reading of our revised manuscript and agree with the nuance they suggest. We revised the wording throughout, including the concluding sentence (now lines 309-313) to clarify that the results provide insights into relative exposure to risk with respect to genomic variability. Our results do suggest that adaptation through gene flow of adaptive alleles from other populations may be a viable mechanism in this system (lines 255-263). We have modified other relevant parts of the Discussion (lines 248-255) to clarify that "when PAI is very high and most adaptive alleles are already or nearly fixed, populations have little potential for future adaptation *with the standing variation they possess*" and that "populations lacking adaptive variability may not be able to compensate for climate change over the next few decades *without the influx of adaptive variation from other populations*". In addition, we have revisited the text and made minor wording changes in a few other places (lines 117, 223-224, 313) to emphasize that we are referring to relative (rather than absolute) vulnerability to climate change. We believe that these revisions address remaining concerns about the wording used to describe inferences based on genomic/geographic offsets and the aspects of climate change risk that can be inferred from these data.